# Genome-Wide Association Study of Airway Wall Thickening in a Korean Chronic Obstructive Pulmonary Disease Cohort

**DOI:** 10.3390/genes13071258

**Published:** 2022-07-15

**Authors:** Ah Ra Do, Do Yeon Ko, Jeeyoung Kim, So Hyeon Bak, Ki Yeol Lee, Dankyu Yoon, Chol Shin, Soriul Kim, Woo Jin Kim, Sungho Won

**Affiliations:** 1Interdisciplinary Program of Bioinformatics, Seoul National University, Seoul 08826, Korea; inspiration@snu.ac.kr; 2Environmental Health Center, Department of Internal Medicine, Kangwon National University, Chuncheon 25948, Korea; dydy7115@naver.com (D.Y.K.); jeeyoung0628@gmail.com (J.K.); 3Department of Radiology, Kangwon National University Hospital, Kangwon National University School of Medicine, Chuncheon 24341, Korea; arsgnm05@naver.com; 4Department of Radiology, Korea University Ansan Hospital, Ansan 15355, Korea; kiylee@korea.ac.kr; 5Department of Chronic Disease Convergence Research, Division of Allergy and Respiratory Disease Research, National Institute of Health, Korea Disease Control and Prevention Agency, Cheongju 28159, Korea; dyoon@korea.kr; 6Institute for Human Genomic Study, College of Medicine, Korea University, Seoul 08826, Korea; chol-shin@korea.ac.kr; 7Department of Internal Medicine, Division of Pulmonary Sleep and Critical Care Medicine, Korea University Ansan Hospital, Ansan 15355, Korea; 8Department of Public Health Sciences, School of Public Health, Seoul National University, Seoul 08826, Korea; 9Institute of Health and Environment, Seoul National University, Seoul 08826, Korea; 10RexSoft Inc., Seoul 08826, Korea

**Keywords:** airway wall thickening, chronic obstructive pulmonary disease, genetic variants, genome-wide association study, single-nucleotide polymorphism

## Abstract

Airway wall thickening (AWT) plays an important pathophysiological role in airway diseases such as chronic obstructive pulmonary disease (COPD). There are only a few studies on the genetic components contributing to AWT in the Korean population. This study aimed to identify AWT-related single-nucleotide polymorphisms (SNPs) using a genome-wide association study (GWAS). We performed GWAS for AWT using the CODA and KUCOPD cohorts. Thereafter, a meta-analysis was performed. Airway wall thickness was measured using automatic segmentation software. The AWT at an internal perimeter of 10 mm (AWT-Pi10) was calculated by the square root of the theoretical airway wall area using the full-width-half-maximum method. We identified a significant SNP (rs11648772, *p* = 1.41 × 10^−8^) located in *LINC02127*, near *SALL1*. This gene is involved in the inhibition of epithelial–mesenchymal transition in glial cells, and it affects bronchial wall depression in COPD patients. Additionally, we identified other SNPs (rs11970854, *p* = 1.92 × 10^−6^; rs16920168, *p* = 5.29 × 10^−6^) involved in airway inflammation and proliferation and found that AWT is influenced by these genetic variants. Our study helps identify the genetic cause of COPD in an Asian population and provides a potential basis for treatment.

## 1. Introduction

Chronic obstructive pulmonary disease (COPD) is one of the most common lung diseases worldwide and is characterized by airflow obstruction that is not fully reversible with treatment [1]. Small airway diseases and parenchymal destruction play a role in the pathogenesis of COPD at different rates over time, resulting in chronic airflow limitation. These pathologies do not always occur simultaneously, and their contribution to the development of COPD differs between individuals [2]. Airway remodeling in asthma and COPD results in airway wall thickening (AWT), which affects lung function. AWT is associated with chronic mucus hypersecretion in larger airways and airway obstruction in smaller airways [3]. The pathological process underlying AWT is chronic inflammation and remodeling of the airway wall by external factors, such as dust [3]. However, there are no certain genetic factors known to influence airway inflammation. Understanding the underlying genetic mechanisms will help develop novel diagnostic techniques and treatment strategies. Many improvements have been made in the quantification of the airway phenotype using computed tomography (CT) quantification methods. We attempted to measure airway wall thickness more objectively using image scanner technology and the associated software.

AWT plays an important pathophysiological role in airway disease [4]. COPD patients with chronic bronchitis have thicker airway walls than COPD patients without chronic bronchitis [5]; AWT plays a major role in COPD, and COPD-associated genes are also associated with AWT [6]. Moreover, two single-nucleotide polymorphisms (SNPs) associated with AWT were identified in male participants from Groningen and Utrecht [3].

However, few studies have explored the genetic mechanism underlying AWT using genome-wide association studies (GWAS) [7]. Therefore, we aimed to identify SNPs associated with AWT through GWAS. This study may enable the identification of genetic mechanisms that aid in understanding the development of AWT.

## 2. Materials and Methods

### 2.1. Study Population

Study participants were chosen from two cohorts: 500 subjects (324 COPD patients and 176 controls) from the COPD in dusty areas (CODA) cohort [8] and 474 subjects (235 COPD patients and 239 matched controls) from the KoGES-Ansan cohort [9,10]. Written informed consent was obtained from all the participants. This study was approved by the Kangwon National University Hospital IRB (KNUH 2012–06-007, clinical trial registration number KCT-0000552) and the Korea University Ansan Hospital IRB (2017AS0070).

### 2.2. Spirometry and Imaging Procedures

Lung function was measured before and after administering 400 μg of salbutamol using EasyOne (NDD, Zurich, Switzerland). The pulmonary function measures were calculated according to the American Thoracic Society and European Respiratory Society guidelines [11]. CT was performed at maximal inspiration and expiration in the supine position using a dual-source CT scanner (Somatom Definition, Siemens Healthcare, Forchheim, Germany for the CODA cohort; Brilliance 64, Philips Healthcare, Cleveland, OH, USA for the KUCOPD cohort). Airways were measured using automatic segmentation software (Aview, Coreline Soft, Seoul, Korea). The AWT at an internal perimeter of 10 mm (AWT-Pi10) was calculated by plotting the square root of the airway wall area against the internal perimeter of each measured airway using the full-width-half-maximum (FWHM) method.

### 2.3. Genotyping and Quality Controls

Genomic DNA was isolated from blood samples. Genotypes were created using the Axiom^™^ Precision Medicine Research Array, which contains more than 860,000 SNPs. Quality control of the SNPs was performed using PLINK [12] and ONETOOL [13]. SNPs were excluded based on the following exclusion criteria: genotype call rate < 95% and the Hardy-Weinberg equilibrium (HWE) test [14], where *p* < 1 × 10^−5^. Subjects were excluded if 0.2 < X chromosome homozygosity < 0.8, genotype call rate < 95%, or heterozygosity rates of SNPs were outside the average heterozygosity rate ± 3 standard deviation (SD). Following the quality control processes, 433 and 387 subjects and 768,913 and 775,371 SNPs were included in the CODA and KUCOPD cohorts, respectively (Figure 1).

### 2.4. Genotype Imputation of SNP Genotype Data

The imputation was conducted using the Michigan imputation server (https://imputationserver.sph.umich.edu, accessed on 20 May 2020) for the CODA and KUCOPD data. We used Haplotype Reference Consortium release v1.1 as a reference panel and considered only ‘not European’ or ‘mixed’ populations [15]. Pre-phasing and imputation were conducted using Eagle V 2.4 and Minimac4, respectively [16,17]. After the imputation processes, the following imputed SNPs were removed: SNPs with Rsq less than 0.3; with duplications; missing genotype rates larger than 0.05; *p* values for HWE less than 1 × 10^−5^; and minor allele frequencies (MAFs) less than 0.05. Additionally, subjects with identity-by-descent > 0.9 and principal component score outside the 5 × IQR_PC_ were excluded. A total of 433 and 387 subjects and 4,941,935 and 4,956,071 SNPs were used for our analyses (Figure 1).

### 2.5. Meta-Analysis of GWAS

GWAS for AWT was applied to CODA and KUCOPD data using linear regression. Airway wall thickness was used as a response variable, and ten PC scores, estimated using PLINK adjusted for population structure, age, and sex, were included as covariates. A meta-analysis of both datasets was performed using METAL program. The sample size of each dataset was used as the weight [18]. The genome-wide significance level was set at α = 5 × 10^−8^.

### 2.6. RNA Expression Association Test

GSE18965 from the GEO database was used to detect significant associations between AWT and the RNA expression of genes identified from the meta-analysis [19]. RNA expression in the airway epithelial cells of nine patients with asthma and seven healthy non-atopic controls was evaluated using the Affymetrix Human Genome U133A Array. The ‘limma’ and ‘balli’ tools of the R package [20,21] were used to detect the differentially expressed genes.

## 3. Results

### 3.1. Subject Characteristics

The demographic properties of the CODA data of 433 subjects and the KUCOPD data of 387 subjects are presented in Table 1. There were significant differences in age, height, weight, and BMI between the CODA and KUCOPD data. Subjects from the CODA data group were much older, shorter, and heavier than those included in the KUCOPD data group. Lung function is indicated by the predicted forced expiratory volume in 1 s measured without a bronchodilator (FEV1.Pred.pre-BD, %); this was lower in the CODA data (83.9 ± 22.7) than in the KUCOPD data (101.7 ± 17.0). Additionally, forced expiratory volume in 1 s divided by forced vital capacity (FEV1/FVC) was lower in the CODA data (65.1 ± 11.5) than in the KUCOPD data (71.2 ± 8.7). There were no significant differences in the level of smoking. In conclusion, there were significant differences in age, physique, and pulmonary functions between both CODA data and the KUCOPD data.

### 3.2. GWAS of Airway Wall Thickness

We performed GWAS with linear regression for the CODA and KUCOPD data. A multi-dimensional scaling plot based on our dataset and the 1000 genome datasets was generated; no evidence of population stratification was found for both the CODA and KUCOPD data (Appendix A). However, both CODA and KUCOPD data were case-control data, and the population stratification was expected to be the main confounder for genetic analyses. Ten PC scores were utilized as covariates to adjust this problem. Quantile-quantile plots for GWAS with CODA data, KUCOPD data, and their meta-analyses are shown in Appendix A. There was no genomic inflation, as indicated by the genomic inflation factors (Λ_CODA_ = 1.00, Λ_KUCOPD_ = 0.99, Λ_meta_ = 1.00). In Appendix A, the top Manhattan plot indicated a genetically significant locus (rs4491106, β = 0.1503, *p* = 1.58 × 10^−8^) in the linear regression of CODA data. However, there were no significant SNPs in the linear regression of KUCOPD data (bottom Manhattan plot). For meta-analysis, rs11648772, located on chromosome 16 in *LINC02127* and Spalt-like transcription factor 1 (*SALL1*), met the genome-wide significance level (*p* = 1.41 × 10^−8^) (Figure 2). Table 2 shows the results for the most significant ten SNPs by meta-analysis. The MAFs of the ten SNPs were compared with those of the SNPs in the Korean reference (Kref) dataset. When there were several genome-wide significant SNPs in the same LD block, only the most significant SNP was included. Figure 3 shows the forest plot for the top ten SNPs in Table 2. In particular, rs11648772, which is located near rs147153117, has been shown to be associated with post-bronchodilator and FEV1/FVC ratio [22]. In summary, we found rs11648772 located near the SALL1 gene is genetically correlated with airway wall thickness in a meta-analysis using CODA and KUCOPD data.

### 3.3. Analysis of Differentially Expressed Genes

To assess the differentially expressed genes associated with AWT, we evaluated six genes (*SALL1*, *CYLD*, *NOD2*, *BRD7*, *ADCY7,* and *HEATR3*) located near rs11648772, using the GSE18965 dataset from the GEO database. *SALL1* gene expression was upregulated in patients with asthma (β_limma_ = 0.12, *p*_limma_ = 0.0173, β_balli_ = 0.12, *p*_balli_ = 0.0092) (Table 3). In short, we confirmed that the expression of the SALL1 gene, the closest gene to the rs11648772, had a positive association with airway wall thickness in asthmatic patients.

## 4. Discussion

In this study, we identified SNPs associated with AWT using genotype and CT data meta-analysis from two Korean cohorts. The most significantly associated SNP was rs11648772, located in *LINC02127* and *SALL1*. *SALL1* was the nearest gene to rs11648772, and its RNA expression was associated with airway wall thickness in asthmatic patients with atopic dermatitis.

Previous studies have assessed airway dimensions, such as lumen area or diameter, or Pi10 with different airway sampling methods, in particular chest CT, in relation to airflow limitation, respiratory symptoms, and emphysema [3]. The high resolution and marked natural contrast between air and normal lung parenchyma enable quantitative measurements using dedicated software. The combination of quantitative parameters offered by these tools enables better analysis of COPD phenotypes and prediction of outcomes. Therefore, in addition to diagnosis, these quantitative measurements allow for the staging of disease severity and phenotyping of patients [23]. In chest CT scan phenotypes, the estimated heritability of both FEV1 and FEV1/FVC is close to 25%, while the heritability of COPD status was estimated to be 37.7% in non-Hispanic whites and African Americans [24].

Identifying SNPs associated with the AWT phenotype through GWAS will enable the discovery of airway disease-related genes. Therefore, we used FWHM to detect SNPs associated with AWT and found that rs11648772 was the most significant in the meta-analysis. rs11648772 is located in the *LINC02127* and *SALL1* genes. *SALL1*, the nearest target gene to rs11648772, encodes a zinc finger transcriptional repressor, which functions in the nucleosome remodeling deacetylase histone deacetylase complex. Asthmatic epithelial cells in children secrete less fibronectin, an important contributor to the dysregulated airway epithelial cell repair. Fibronectin is an essential component of the provisional extracellular matrix because it provides a surface for epithelial migration and proliferation. Impaired fibronectin expression contributes to the abnormal epithelial repair seen in asthmatic airways [19]. *SALL1* inhibits cell migration by preventing epithelial–mesenchymal transition (EMT) and downregulating the expression of stem cell markers [25]. Furthermore, it acts as a tumor suppressor by inhibiting Wnt/β-catenin signaling [25]. Recently, the role of EMT in airway remodeling was established [26]. In COPD, the airway epithelium may be damaged and/or activated by irritants, such as the constituents of cigarette smoke, stimulating the deposition of collagen from myofibroblasts in the lamina propria [27]. *SALL1* influences EMT inhibition in glial cells, and it could affect bronchial wall depressions in asthmatic patients. Therefore, we hypothesize that *SALL1* is associated with the EMT pathway and inhibition of cell proliferation in the bronchial airway. However, further studies on *SALL1* expression in patients with bronchial damage are essential to elucidate its role in bronchial epithelial cells, and the genetic functions of rs11648772 need further analysis.

In addition to *SALL1*, rs11970854 and rs16920168 were involved in airway inflammation and proliferation; rs11970854 is located near *KLRG2* (dist = 7,721) on chromosome 7, and rs16920168 is located in the *LYN* of chromosome 1. *KLRG2* is associated with the inflammatory Innate counterpart of type 2 helper T cells (ILC2s); it expresses GATA3, a key transcription factor of ILC2 [28]. Increased ILC2 levels cause pathogenic chronic inflammation and/or alterations in the structure, repair, and developmental processes of the lung [29]. *LYN* downregulates allergen-induced airway inflammation, and its overexpression decreases mucus secretion and *MUC5AC* transcription in mice exposed to allergens [30].

In a cohort recruited from the Dutch NELSON, novel AWT-associated SNPs rs10794108 and rs7078439 on the *C10ORF90* and *DOCK1* loci were identified. These SNPs were not significant in the meta-analysis and KUCOPD data; however, in CODA data, rs10794108 was replicated with *p* = 0.04352 in the linear regression. We also found a significant SNP (rs4492106) located on chromosome 5 in the CODA data group. 

This study has some limitations. We could not identify the SNPs reported in a previous study based on the European population. This could be attributed to ethnic differences. Although our analysis was conducted in two independent groups, the small sample size could decrease the statistical power. Therefore, further studies with larger population sizes are needed.

To conclude, we identified SNPs associated with the AWT phenotype, which could influence the pathogenesis of airway diseases and provide a potential basis for treatment in Asian populations.

## Figures and Tables

**Figure 1 genes-13-01258-f001:**
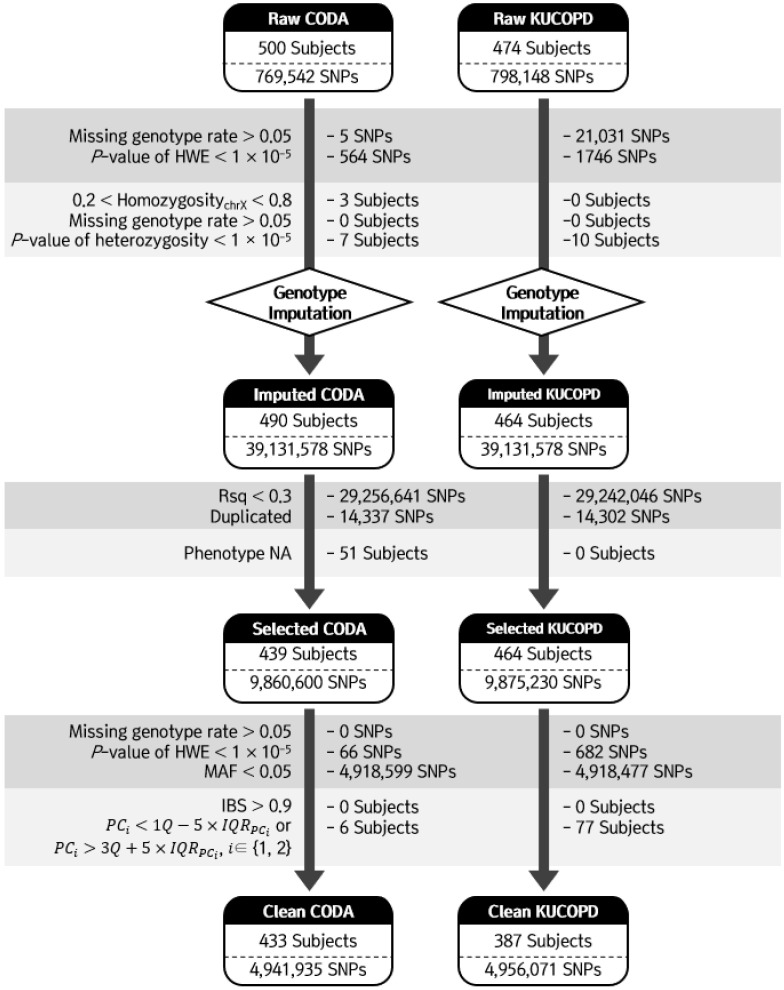
Workflow of quality control for CODA and KUCOPD data. Several standard quality control steps were produced for CODA and KUCOPD data to exclude outlier single-nucleotide polymorphisms (SNPs) and subjects.

**Figure 2 genes-13-01258-f002:**
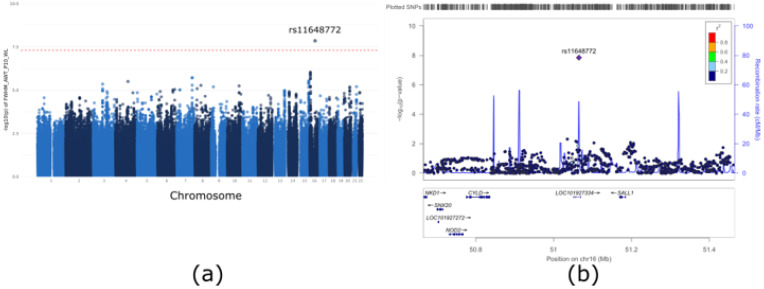
Meta-analysis results. (**a**) Manhattan plot of *p* values from the genome-wide association study (GWAS). One region on chromosome 16 met genome-wide significance (*p* < 5 × 10^−8^) according to the meta-analysis of CODA and KUCOPD data. (**b**) The expanded Manhattan plot of the 800 kb region shows both genotyped and imputed single-nucleotide polymorphisms (SNPs). rs11648722, the most significant SNP, is indicated by the purple diamond, and other SNPs are coloured according to their r^2^ values in relation to that of rs11648722. *LOC101927334* (*LINC02127*) and *SALL1* are the closest located genes.

**Figure 3 genes-13-01258-f003:**
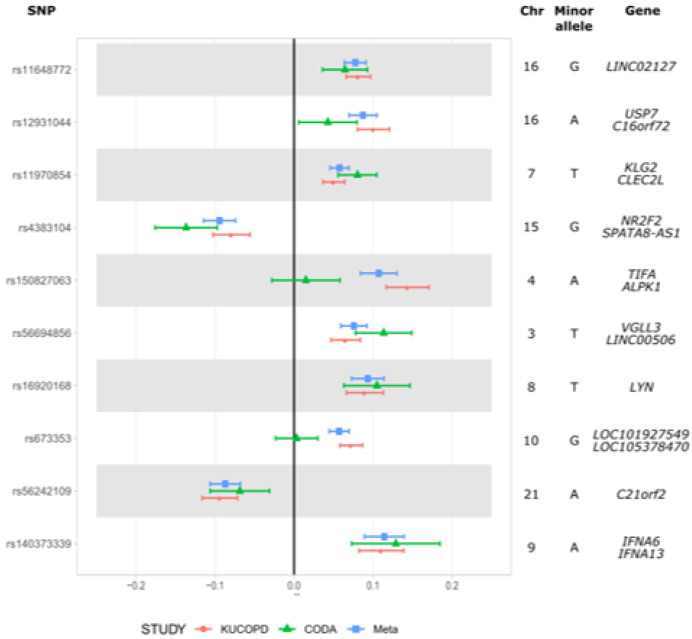
Forest plot for the top ten SNPs associated with airway wall thickness.

**Table 1 genes-13-01258-t001:** Descriptive statistics.

Variable	CODA Cohort (*n* = 433)	KUCOPD Cohort (*n* = 387)	Total (*n* = 820)	*p* Value
Female sex, *n* (%)	118 (27.3%)	96 (24.8%)	214 (26.1%)	0.4737
Age	72.0 ± 7.1	63.0 ± 7.6	67.7 ± 8.6	<0.001
Height	159.3 ± 9.3	164.1 ± 7.7	161.5 ± 8.9	<0.001
Weight	59.9 ± 10.2	65.7 ± 9.5	62.7 ± 10.3	<0.001
BMI	23.6 ± 3.2	24.4 ± 2.9	24.0 ± 3.1	<0.001
Pack-Year	17.3 ± 22.9	16.6 ± 20.3	17.0 ± 21.7	0.8058
Smoking, *n* (%)				0.4748
Never	156 (36.2%)	152 (39.4%)	308 (37.7%)	
Former	179 (41.5%)	160 (41.5%)	339 (41.5%)	
Current	96 (22.3%)	74 (19.2%)	170 (20.8%)	
FEV1. Pred. pre-BD (%)	83.9 ± 22.7	101.7 ± 17.0	92.0 ± 22.1	<0.001
FEV/FVC (%)	65.1 ± 11.5	71.2 ± 8.7	67.9 ± 10.7	<0.001
COPD, *n* (%)	278 (64.2%)	194 (50.1%)	472 (57.6%)	<0.001
FWHM_AWT_P10_WL	4.7 ± 0.4	4.4 ± 0.2	4.5 ± 0.3	<0.001

Mean ± standard deviation (SD) values are shown in each cell. BMI: body mass index; EI: emphysema index; FEV1: forced expiratory volume in one second; BD: bronchodilator; FVC: forced vital capacity; FWHM_AWT_P10_WL: AWT-Pi10 (mm) in the whole lung by full-width-half-maximum methods.

**Table 2 genes-13-01258-t002:** Meta-analysis results.

Chr	BP	SNP	MAF_KRef_	Effect	SE	*p* Value	Alt /Ref	CODA Cohort			KUCOPD Cohort			Gene
								*p* Value	MAF	HWE	*p* Value	MAF	HWE	
16	51065076	rs11648772	0.19	−0.08	0.01	1.41 × 10^−8^	G/T	0.0240	0.24	0.6950	2.87 × 10^−7^	0.25	0.4168	*LINC02127*
16	9109561	rs12931044	0.13	0.09	0.02	9.22 × 10^−7^	A/G	0.2466	0.13	1.0000	1.05 × 10^−6^	0.13	0.5096	*USP7* (dist = 52,220), *C16orf72* (dist = 75,976)
7	139176178	rs11970854	0.45	0.06	0.01	1.92 × 10^−6^	T/C	0.0012	0.47	0.9235	3.35 × 10^−4^	0.49	0.9190	*KLRG2* (dist = 7721), *CLEC2L* (dist = 32,496)
15	97215196	rs4383104	0.10	0.09	0.02	3.02 × 10^−6^	G/A	0.0006	0.12	0.2442	8.28 × 10^−4^	0.09	0.3565	*NR2F2* (dist = 331,704), *SPATA8-AS1* (dist = 100,039)
4	113212158	rs150827063	0.10	0.11	0.02	3.15 × 10^−6^	A/G	0.7282	0.09	0.7667	2.07 × 10^−7^	0.07	0.7092	*TIFA* (dist = 5099), *ALPK1* (dist = 6341)
3	87115233	rs56694856	0.14	0.08	0.02	4.45 × 10^−6^	T/C	0.0015	0.15	0.5605	5.12 × 10^−4^	0.17	0.5816	*VGLL3* (dist = 74,960), *LINC00506* (dist = 23,197)
8	56895276	rs16920168	0.10	0.09	0.02	5.29 × 10^−6^	T/C	0.0127	0.10	0.2902	1.58 × 10^−4^	0.09	0.3644	*LYN*
10	107851568	rs673353	0.33	−0.06	0.01	5.68 × 10^−6^	G/T	0.8980	0.33	0.8277	6.05 × 10^−7^	0.33	0.2038	*LOC101927549* (dist = 271,477), *LOC105378470* (dist = 48,355)
21	45757631	rs56242109	0.13	−0.09	0.02	6.27 × 10^−6^	A/G	0.0687	0.12	0.5001	3.77 × 10^−5^	0.12	0.0962	*C21orf2*
9	21359818	rs140373339	0.03	0.11	0.03	6.41 × 10^−6^	A/C	0.0220	0.06	0.6320	1.18 × 10^−4^	0.06	0.1563	*IFNA6* (dist = 8932), *IFNA13* (dist = 7553)

Chr: chromosome; BP: base pair; SNP: single-nucleotide polymorphism; SE: standard error; MAF: minor allele frequencies; HWE: Hardy-Weinberg equilibrium; KRef: Korean reference data used in this study were provided by the CODA (http://coda.nih.go.kr, accessed on 31 May 2020). CODA accession numbers R002754, R002755, R002774, R002814 and R002854.

**Table 3 genes-13-01258-t003:** Differentially expressed genes analysis from GSE18965.

Gene	Chr	BP	Coefficient		*p* Value	
			limma	balli	limma	balli
*SALL1*	16	51,169,886–51,185,278	0.12	0.12	0.0173	0.0092
*CYLD*	16	50,775,961–50,835,846	0.02	0.84	0.8502	0.9079
*NOD2*	16	50,727,514–50,766,988	0.03	0.64	0.6576	0.7697
*BRD7*	16	50,347,398–50,402,845	0.01	0.93	0.9312	0.9609
*ADCY7*	16	50,280,048–50,352,046	−0.01	0.93	0.9764	0.9864
*HEATR3*	16	50,099,852–50,140,298	0.11	0.33	0.3587	0.5156

Chr: chromosome; BP: base pair.

## Data Availability

Not applicable.

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
