# Peer review of "Genome-Wide Association Study of Airway Wall Thickening in a Korean Chronic Obstructive Pulmonary Disease Cohort"

_genes, 2022, doi:10.3390/genes13071258_

Round 1

Reviewer 1 Report

The title of the article fully reflects the content of the article.

The Abstract section contains the necessary information for the reader: the pathophysiological role of thickening of the walls of the respiratory tract (AWT) in chronic obstructive pulmonary disease (COPD) is indicated, the small development of the problem of genetic components contributing to AWT in the Korean population, the purpose of the study, the main research methods, the results of the study. The conclusions are clear and consistent with the results of the study. The study helps to identify the genetic cause of COPD in the Asian population and provides a potential basis for treatment.

All keywords are necessary and reflect the research topic presented by the authors.

In the Introduction section, the authors presented a general understanding of the prevalence, symptoms and pathogenesis of COPD. The important role of airway remodeling in COPD, as well as the pathogenetic factors underlying it, is noted. A small number of studies of the genetic mechanisms underlying AWT have been noted. Ultimately, the authors hope that the present study may allow the identification of genetic mechanisms that help in understanding the development of AWT.

The purpose of the study is clear. The connection between the articles cited in the "Introduction" and the purpose of this study is visible.

In the section "Materials and methods", the cohorts of the study participants are presented quite fully and correctly. Very important information about the written informed consent of the study participants and the approval of the IRB Hospital of Kangwon National University and the IRB Hospital of Ansan Korean Research University. The procedures of spirometry and visualization, as well as Genotyping and quality control, Genotype imputation according to SNP genotype data, GWAS Meta-analysis, RNA Expression Association Test are fully and clearly presented. All the information in the section "Materials and Methods" is extensive and necessary. The design of the study is clear. In this section, the authors make references to previously conducted work, which is necessary to understand the researchers' intention.

The section "Results" presents the main results of the study in separate chapters: 3.1. Characteristics of the subject, 3.2. GWAS of the wall thickness of the respiratory tract, 3.3. Analysis of differentially expressed genes. It is recommended to supplement individual sub-chapters with short conclusions.

All figures and tables are legible, necessary and complement the content of the article. All the tasks planned by the authors have been completed.

In the "Discussion" section, using the literature, the authors analyzed the results obtained. The article contains certain limitations: for example, the authors in the Korean population were unable to identify SNPs reported in a previous study based on the European population. This could be explained by ethnic differences. In addition, the small sample size may have reduced statistical reliability. However, the authors note the need for further studies with larger populations. This is certainly the right way and will improve the understanding of the pathophysiological mechanisms of COPD and will provide a potential basis for the treatment of the disease in Asian populations.

The conclusion is correct and follows from the results of the conducted research.

The submitted manuscript does not cause any concerns. The manuscript does not cause any ethical problems. All references to publications presented by the authors in the article are necessary and correct, made in the right style.

I have no concerns about the similarity of this article with other articles published by the same authors.

Reviewer 2 Report

The objective of this study was to identify single-nucleotide polymorphisms (SNPs) associated with airway wall thickening using a genome-wide association study. 

The description of study population was confusing. In Line 72 - 74, it was mentioned that "500 subjects from the COPD in dusty areas (CODA) cohort, and 240 COPD patients and 240 matched controls (KUCOPD) from the KoGES-Ansan cohort". Were all 500 subjects from the CODA cohort patients with COPD?

Were the genotyping results for COPD patients and matched controls from KUCOPD cohort separated when the authors performed the meta-analysis of the genotyping data? 

I did not find the information on how the DNA samples were obtained? From the blood samples or bronchial biopsy specimens?

Round 2

Reviewer 2 Report

Regarding the authors' Response 2, what is the rationale for considering the population stratification to be the main confounder for genetic analyses.  Since it was a GWAS of airway wall thickening in patients with COPD, it would be reasonable to examine COPD as another main confounder for genetic analyses. Have the authors ever tried to separate the control subjects and COPD patients and then compare the analysis results between the two groups? Was any significant difference identified between control and COPD groups?